# The antiaging effects of a product containing collagen and ascorbic acid: *In vitro, ex vivo*, and pre-post intervention clinical trial

Tae Kyeong Ryu[1,2,3⊙], Hanna Lee[1,2,3⊙], Dong Keon Yon[4], Da Yeong Nam[1,2,3], Soo Yun Lee[1,2,3], Byung Ho Shin[1,2,3], Go Woon Choi[1,2,3], Da Som Jeon[1,2,3], Bo Bae Oh[1,2,3], Ji Hyun Kim[1,2,3], Young Yoon[1,2,3], Hyun Jeong Kim[1,2,3], Luc Duteil[5], Christelle Bruno-Bonnet[6], Chan Yeong Heo[1,2,3]*, So Min Kang[1,2,3]*

1 Department of Plastic and Reconstructive Surgery, Seoul National University Bundang Hospital, Seongnam, South Korea, 2 Korean Skin Research Center, Seongnam, South Korea, 3 H&BIO Corporation/R&D Center, Seongnam, South Korea, 4 Center for Digital Health, Medical Science Research Institute, Kyung Hee University College of Medicine, Seoul, South Korea, 5 Centre of Clinical Pharmacology Applied to Dermatology (CPCAD), Hôpital l'Archet 2, Nice, France, 6 Weishardt International Group, Rond-Point Georges Jolimaître, Graulhet, France

⊙ These authors contributed equally to this work.
* lionheo@gmail.com (CYH); doctork721@koreansrc.com (SMK)

**Data Availability Statement:** All relevant data are within the article and its Supporting information files.

## Abstract

Various substances, including collagen (Naticol®) and ascorbic acid, that inhibit and prevent skin aging have been studied. Collagen prevents skin aging, has anti-inflammatory effects, and assists in normal wound healing. Ascorbic acid is a representative antioxidant that plays a role in collagen synthesis. To achieve a synergistic effect of collagen and ascorbic acid on all skin types, we prepared a product named "TEENIALL." In addition, we used a container to separate ascorbic acid and collagen to prevent the oxidation of ascorbic acid. To confirm the effects of TEENIALL, we first confirmed its penetrability in fibroblasts, keratinocytes, melanocyte, and human skin tissues. Thereafter, we confirmed the collagen synthesis ability in normal human fibroblasts. Based on the results of *in vitro* tests, we conducted a clinical trial (KCT0006916) on female volunteers, aged 40 to 59 years, with skin wrinkles and hyperpigmentation, to evaluate the effects of the product in improving skin wrinkles, skin lifting, and pigmentation areas before using the product, and after 2 and 4 weeks of using the product. The values of nine wrinkle parameters that were evaluated decreased and those for skin sagging, pigmentation, dermal density, and mechanical imprint (pressure) relief were improved. Skin wrinkle and pigmentation were evaluated to ensure that the improvement effect was maintained even after 1 week of discontinuing the product use. The evaluation confirmed that the effects were sustained compared to those after 4 weeks of using the product. Additionally, skin wrinkles, skin lifting, radiance, and moisture content in the skin improved immediately after using the product once. Based on the results of *in vitro* and *ex vivo* experiments and the clinical trial, we show that the product containing ascorbic acid and collagen was effective in alleviating skin aging.

**Funding:** The author(s) received no specific funding for this work.

**Competing interests:** The authors have declared that no competing interests exist.

## Introduction

The skin undergoes a wide variety of physiological and pathological changes over time [1]. In view of the increasing interest in beauty attributes in recent times, besides issues of physiological and pathological changes in the skin, studies are being conducted to decipher the mechanisms underlying skin aging, and to find the associated biological changes, which might help in devising strategies for inhibition of the skin aging [2,3].

Skin aging can be largely categorized as intrinsic aging, which is inevitable over time, and extrinsic aging (photoaging), which is caused due to a variety of reasons, such as exposure to sunlight, stress, and smoking; the phenotypes of the two aging types differ [4]. Intrinsic aging is relatively mild, and is manifested as fine lines, dry skin, and decreased elasticity [5]. However, typical photoaging results in thick and deep wrinkles, pigmentation, very rough texture, and sagging [6]. With the accumulation of damage, there is an increase in the levels of reactive oxygen species (ROS) production [7]. Moreover, there are changes in the properties and amounts of matrix proteins, as the signaling system is regulated by changes in the levels of cytokines [8,9].

Wrinkles are distinct clinical features of aging that appear with the decrease in collagen and elasticity due to an increase in matrix metalloproteinases (MMPs), such as collagenase and elastase; histologically, water loss, epidermal contraction, and flattening are seen [10]. In the dermis, collagen accounts for more than 90%, and type I and type III collagen account for 85–90% and 10–15% of the total collagen present, respectively [11,12]. Another clinical feature—pigmentation—is caused due to the increase in melanin content through activation of melanocytes [13]. Various substances with potential to inhibit and prevent such skin aging are being studied. Among these are collagen and ascorbic acid. Ingestion or topical application of collagen prevents skin aging, has anti-inflammatory effects, and assists in normal wound healing [2,3]. Weishardt, a world-renowned collagen manufacturer, has a variety of fish collagen peptides, named Naticol®, with different molecular weights and formulations; these peptides are widely used in cosmetics, medicines, and food [14]. In particular, Naticol® BPMG is composed of collagen type 1 and type 3, which are the main components of the dermis. It is produced from high quality raw materials (fish skin) and through specific enzymatic hydrolysis [15].

Ascorbic acid is a representative water-soluble antioxidant. Two different mechanisms are described for the antioxidant properties of ascorbic acid [16]. First, it reacts with free radicals in aqueous components of the body, such as cytoplasm, plasma, and extracellular fluid, and inactivates them [17]. It lowers the activity of lipoxygenase, an enzyme that catalyzed the peroxidation of lipids. Second, it regenerates the oxidized form of vitamin E, which prevents lipid peroxidation very effectively [18]. It also plays a role in collagen synthesis [19]. For the synthesis of collagen, proline and lysine must be converted to hydroxyproline and hydroxylysine to enable stronger hydrogen bonding between collagen chains. Three enzymes (prolyl-3-hydroxylases, prolyl-4-hydroxylases, and lysyl hydroxylases) use ascorbic acid as a coenzyme [20]. The antiaging effects of ascorbic acid in clinical practice have been shown in many previous studies [21].

This study was conducted to evaluate the effects of applying TEENIALL (TEENIALL Double Concentrate Vitamin C Youth Ampoule, WELBORN, Korea) containing Naticol® (Weishardt, France) and ascorbic acid (Quali®-C, United Kingdom) together as an antiaging compound. First, experiments were conducted using cells and tissues to confirm the permeability and collagen synthesis ability of Naticol® and ascorbic acid, and thereafter, clinical studies were conducted on human subjects. There is a limit for consumption of ascorbic acid owing to the stability problem associated with its oxidation. To overcome this, an effort was

made by using a special container. Before opening the product, ascorbic acid and collagen were separated, and the two substances could be mixed immediately before use. Moreover, the container was designed to block light. We evaluated the synergistic effect of Naticol® and ascorbic acid and the benefits of using the special container on the antiaging effect. We also assessed whether the effect was maintained even after the discontinuation of the product. The effect was verified through various clinical tests on 21 Korean female volunteers, aged 40–59 years, with skin folds (eye corners, forehead, and nasolabial folds) and pigmentation on the face (Figs 1 and 2A).

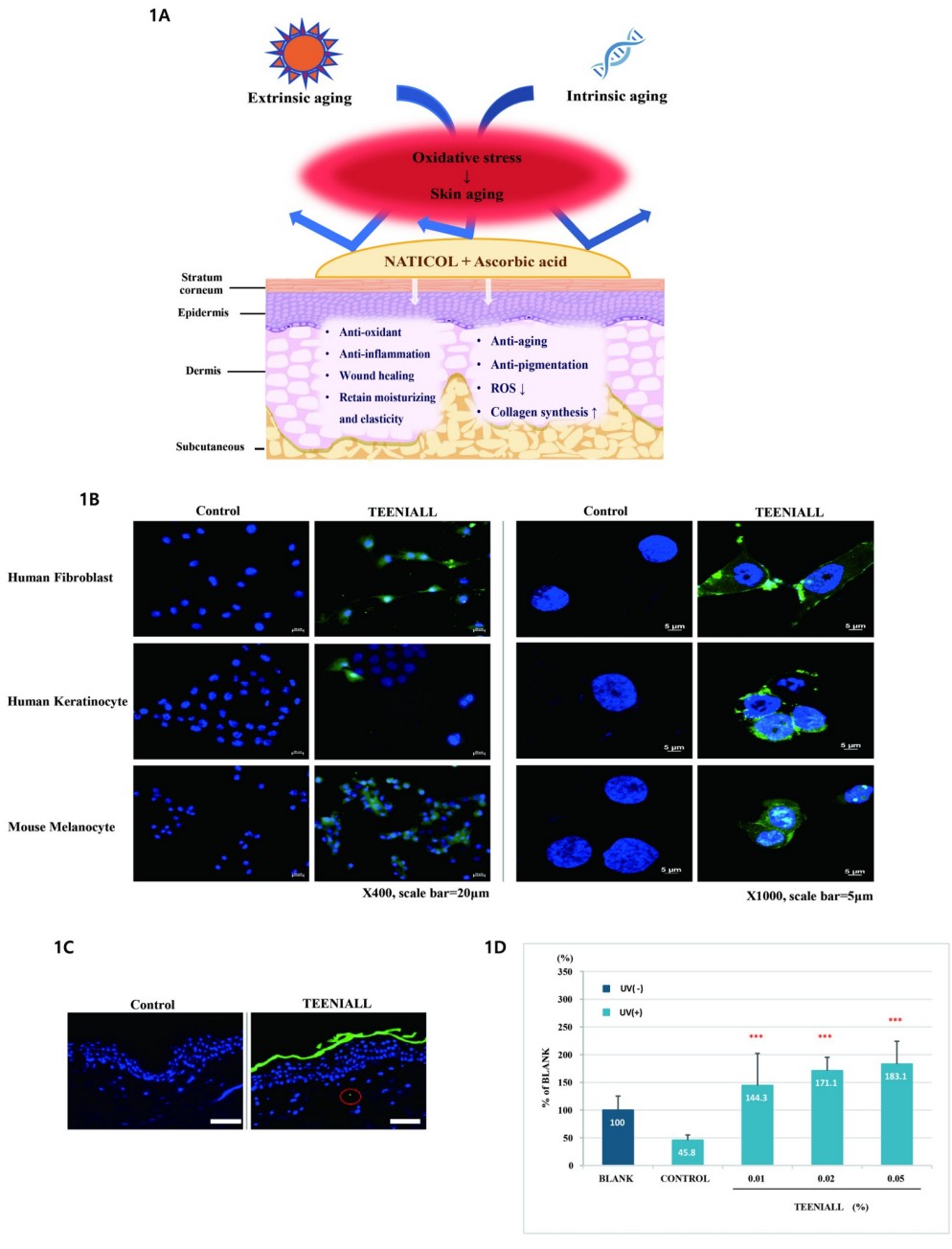

**Fig 1. CONSORT 2010 flow diagram.**

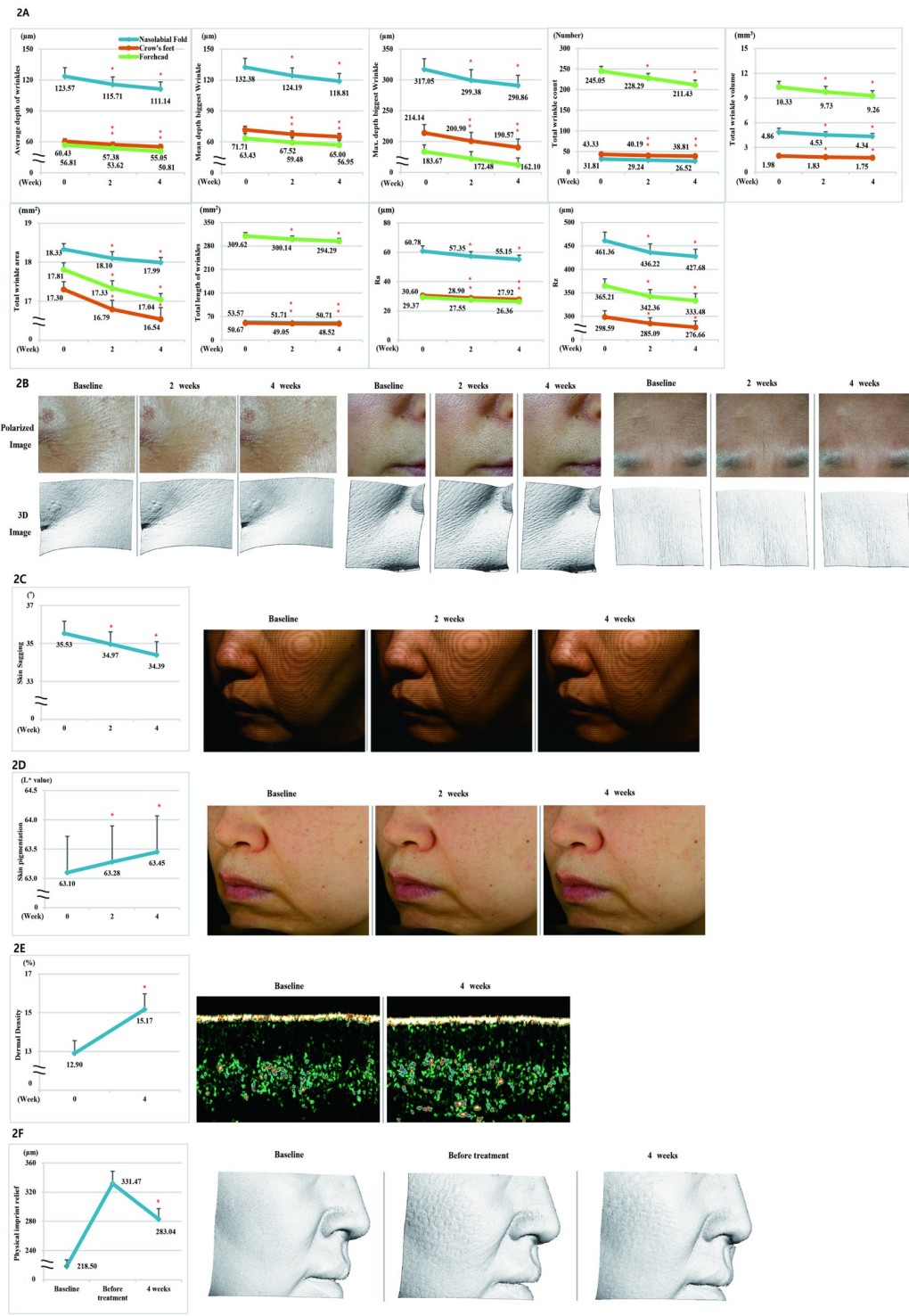

**Fig 2.** (A) Collagen and ascorbic acid have beneficial effect on skin aging. (B) Fluorescence imaging of fibroblasts, keratinocytes, and melanocytes in the control and test groups at ×400 (Scale bar = 20 μm) and ×1000 (Scale bar = 5 μm) magnification. (C) fluorescence imaging of a cross-section of human skin tissue in the control and test groups (×400, Scale bar = 50 μm, n = 20). (D) Analysis of collagen synthesis depending on the concentration of the test product (n = 4, ***p < 0.001).

## Materials and methods

### Test product

TEENIALL (TEENIALL Double Concentrate Vitamin C Youth Ampoule, WELBORN, Korea) is composed of powder type ascorbic acid and liquid type collagen (Naticol BPMG). Ascorbic acid contained in this product is in the form of fine particles, with a diameter of 37 μm, and the molecular weight of collagen is 1.74 Mw/Mn. In clinical trial, this product was divided into powder and liquid components, so that both formulations could be mixed and applied immediately before use.

### *In vitro* and *ex vivo* analyses

**Cell penetration assay.** The cell penetration assay was performed using human fibroblasts (Hs68 cell line), human keratinocytes (HaCaT cell line), and mouse melanocytes (B16F10 cell line). Cells were cultured in Dulbecco's modified Eagle's medium (DMEM; Gibco Invitrogen, Carlsbad, USA), containing the TEENIALL powder. Thereafter, the cells were stained with DAPI (VECTASHIELD$^®$ Antifade Mounting Medium with DAPI). Fluorescence of cells was measured using a fluorescence microscope (Zeiss Axio Observer 7, Germany).

**Measurement of collagen synthesis.** Human normal fibroblasts (neonatal; Gibco, C0045C) were treated with DMEM containing TEENIALL at a concentration of 0.01, 0.02, or 0.05%. Collagen synthesis was measured using an ELISA kit (Procollagen Type I C-peptide [PIP] EIA Kit, TaKaRa, Tokyo, Japan).

**Penetration assay.** The penetration assay was performed using Franz diffusion cells with a diffusion area of 3 cm$^2$, and a 7 mL receptor chamber volume. The receptor medium contained saline solution. TEENIALL (0.4 mL), with labeled protein (Alexa Fluor™ 488, Invitrogen, USA) and control solution (PBS), was applied to the surface of the human skin tissue (Derma: Lab™, Korea). Thereafter, tissue slides were made and the sections were stained with DAPI. Fluorescence from skin sections was measured using a fluorescence microscope.

### Clinical trial

**Study procedure.** *Sample size calculation.* Although there has been no study on the antiaging potential of TEENIALL (ascorbic acid and collagen), prior to conducting clinical research, we calculated the sample size based on a our *in vivo* study on the relationship between collagen synthesis and TEENIALL [22].

We originally calculated that for each group to have a 95% power and 0.05 alpha (sample size calculation) to report a 44.5%–83.1% (18.7% in certain experiments) improvement in collagen synthesis (229.1 [SD 31.5] ng/mL for control group versus 269 ng/mL in certain experiments for treatment group), at a 5% significance level, we would need to enroll 16 patients in each group. Considering the various exclusion conditions (30%), finally, we included 21 participants [23,24].

This study was conducted in accordance with Good Clinical Practice (GCP), Ministry of Food and Drug Safety (MFDS), and Standard Operating Procedures (SOP) of KSRC CO., LTD. The study was approved by the Institutional Review Board at H&BIO KSRC CO. (IRB: HBABN01-210315-HR-0205-01, CRIS number: KCT0006916). Over 20 subjects meeting the inclusion and exclusion criteria were enrolled. For 4 weeks, the product was applied to subjects on a daily basis around the whole face after washing the face twice a day. In this study, treatment was applied for 4 weeks; the study was conducted as a single test and the measurements were made immediately after the application of the product once, and 2 and 4 weeks later. The study investigator explained the purpose, procedures, and anticipated adverse reactions or side

effects to the subjects. The investigator received an informed consent form (ICF) and case report form (CRF) from the subjects who agreed to participate in the study prior to it. The study was conducted from March to June 2021.

Over 20 subjects were recruited according to the MFDS guidelines. For 4 weeks, the product was applied to subjects on a daily basis over the entire face after washing the face twice a day. The application was discontinued for 1 week. The inclusion and exclusion criteria were set as follows:

## Inclusion criteria

- Korean female volunteers, aged 40–59 years, with skin folds (eye corners, forehead, and nasolabial folds) and pigmentation on the face.

- Subjects who were not hypersensitive to vitamin C.

- Healthy subjects, free from acute and chronic diseases, including skin conditions.

- Subjects who had voluntarily signed the informed consent form after understanding the complete explanation of the purpose and protocol of this study.

- Subjects who were available for follow up during the study period.

## Exclusion criteria

- Women who were pregnant, lactating, or planning to become pregnant within 6 months.

- Subjects who had skin diseases, including active atopic dermatitis, psoriasis, eczema, and active seasonal allergies on the test region.

- Subjects who had used antibacterial agents, immunosuppressants, or external skin preparations containing steroids and treatments for chronic skin conditions for more than 1 month to treat skin conditions on the test region.

- Subjects who had not passed 1 month since participating in the same study.

- Subjects who used the same or similar efficacy cosmetics and medicines on the test site within 3 months prior to the start of the study.

- Subjects with chronic diseases (asthma, diabetes, hypertension, etc.).

- Subjects on contraceptives, antihistamines, and anti-inflammatory drugs.

- Subjects who were employed in this clinical research institute.

- Subjects who were considered inappropriate according to the judgment of the investigator.

Twenty-one subjects (females; mean age, 50.95 ± 5.54 years; age, 40–59 years) participated in the 4-weeks treatment test (two of the twenty-three subjects dropped out [follow-up failure: #03 and #09]), and twenty subjects (mean age, 51.40 ± 5.28 years; age, 40–59 years) participated in the discontinuation 1 week test after 4-weeks treatment (one more subject dropped out [follow-up failure: #11]). Subjects who met the inclusion and exclusion criteria completed the study according to the protocol. The information and characteristics of the subjects were investigated using questionnaires.

**Measurement of wrinkles.**   The Primos-CR (Canfield Scientific, Parsippany-Troy Hills, New Jersey) was used to evaluate the three-dimensional surface of the skin. Data files captured using the Primos-CR were analyzed using the Primos software.

**Measurement of skin sagging.**  F-RAY (BEYOUNG, Korea) images of facial contour curves of the selected cheek area were taken before and immediately after the use of the product. The angle of the contour line was analyzed for the captured contour image using an analysis program, Image Pro® 10 (Media Cybernetics, USA).

**Measurement of pigmentation.**  The skin color of the cheek of volunteers was measured using a CM-26dG spectrophotometer (Konica Minolta, INC., Osaka, Japan). The $L^*$ value expresses the relative brightness following the Commission Internationale de l'Eclairage (CIE) $L^*a^*b^*$ system.

**Measurement of dermal density.**  The skin dermal density of the cheek of volunteers was measured using the Ultrasound Probe of DermaLab® Series SkinLab Combo (Cortex Technology, Denmark). Low density was displayed as a dark color and high density was displayed as a bright color based on the signal strength.

**Measurement of mechanical imprint (pressure) relief.**  The skin pressure impression of the cheek of volunteers made by applying physical force to uneven cotton (8room, Korea). The Primos-CR was used to evaluate the surface of the skin. The image captured using the Primos-CR was analyzed using the Primos software.

**Measurement of skin gloss (radiance).**  Mark-Vu (Mark-Vu®; PSI PLUS Co., Ltd., Suwon, Korea), a skin diagnostic imaging system, was used to analyze the skin gloss. The analysis was performed in the designated area on the polarized image using the Image-Pro® 10 (Media Cybernetics, USA) program.

**Measurement of hydration in the stratum corneum.**  Dermal hydration was measured dielectrically with an open-ended coaxial probe that was layered in structure (Moisture Meter-D; Delphin Technologies Ltd.).

**Statistical analysis.**  Statistical analysis was performed using the SPSS® software program (IBM, USA). The normality was verified using the Shapiro–Wilk test and by examining the kurtosis and skewness. Statistical analysis of variables for parametric values was performed using the paired *t*-test and RM-ANOVA. Two-side *p*-values $<0.05$ were considered significant (*p*-values $<0.05$; * and *p*-values $<0.001$; ***).

## Results

### *In vitro* and *ex vivo* analyses

**Cell penetration assay.**  Significant fluorescence from many small granules in the cytoplasm was observed in the test group of fibroblasts, keratinocytes, and melanocytes indicating the penetration of the product (TEENIALL) into these cells (Fig 2B, S1 Table).

**Skin penetration assay.**  We conducted a skin stratum corneum permeability test to confirm the effectiveness of collagen and ascorbic acid permeation in the stratum corneum (n = 20). Skin layer fluorescence was observed in 50% of the randomly photographed areas in the experimental group unlike in the control group. Fluorescence of the test substance was observed in the stratum corneum and human skin tissue, which indicates that the test substance penetrated the stratum corneum (Fig 2C).

**Measurement of collagen synthesis.**  We investigated whether collagen and ascorbic acid affect the biosynthesis of collagen. As evident from the results of the collagen production test in normal human fibroblasts, the expression of collagen decreased by about 54.2% upon UV irradiation. In addition, comparing the change in the amount of collagen in the case of test product and that in the group without UV irradiation (BLANK; No UV irradiation and treatment), the loss in collagen expression due to UV radiation was completely recovered to 144.3% for 0.01% treatment, 171.1% for 0.02% treatment, and 183.1% for 0.05% treatment (Fig 2D, S1 Table).

## Analyses after treatment for 4 weeks

The results of the clinical trial showed changes in wrinkles around the eyes, nasolabial folds, and forehead; skin lift; reduction in skin pigmentation, dermal density, and mechanical imprint (pressure) relief after 4 weeks of using the product.

**Analysis of wrinkles (Crow's feet, nasolabial fold, and forehead).** We conducted a test on 21 subjects to confirm the antiwrinkle effect of the product. When compared with the baseline, all parameters of skin wrinkles (Average depth of wrinkles, Mean depth biggest wrinkle, Max. depth biggest wrinkle, Total wrinkle count, Total wrinkle volume, Total wrinkle area, Total length of wrinkles, Ra [Arithmetic average], and Rz [Average maximum height of the profile]) improved significantly at 2 and 4 weeks after treatment ($p < 0.05$); for Crow's feet, the rate of change in the abovementioned nine all parameters was 8.90%, 9.36%, 11.01%, 10.43%, 11.62%, 4.39%, 4.24%, 8.76%, and 7.34%, respectively, after 4 weeks compared with the baseline; for nasolabial fold, the rate of change in the parameters was 10.06%, 10.25%, 8.26%, 16.63%, 10.70%, 1.85%, 5.34%, 9.26%, and 7.30%, respectively, after 4 weeks compared with the baseline; for forehead, the rate of change in the parameters was 10.56%, 10.22%, 11.74, 13.72%, 10.36%, 4.32%, 4.95%, 10.25, and 8.69%, respectively, after 4 weeks compared with the baseline (Fig 3A and 3B, S1 Table).

**Analysis of skin sagging.** We conducted a test on 21 subjects to confirm the skin-lifting effect of the product. When compared with the baseline, skin sagging was significantly improved ($p < 0.05$) at 4 weeks after the treatment, with a 3.21% change (Fig 3C, S1 Table).

**Analysis of skin pigmentation.** We investigated the effect of the product in improving skin pigmentation in 21 subjects. When compared with the baseline, skin pigmentation was significantly improved ($p < 0.05$) at 4 weeks after the treatment, with a 0.55% change (Fig 3D, S1 Table).

**Analysis of dermal density.** We conducted a study on 21 subjects to evaluate the effect of the product in improving dermal density. When compared with the baseline, dermal density was significantly improved ($p < 0.05$) at 4 weeks after the treatment, with a 17.60% change (Fig 3E, S1 Table).

**Analysis of mechanical imprint (pressure) relief.** We conducted a test on 21 subjects to evaluate the effect of the product on the mechanical imprint (pressure) relief. When compared with the baseline, the value of the parameter before the treatment was 51.7% and that after 4 weeks treatment was 29.54%. Thus, after 4-weeks treatment the parameter improved by 22.16% compared with that before the treatment (Fig 3F, S1 Table).

## Analysis after 1 week of discontinuation of the product

After observing the improvement effects after 4 weeks of the product use, we further checked the sustenance of the effects after 1 week of discontinuation of the product use.

**Analysis of Crow's feet.** We conducted a study on 20 subjects to evaluate the improvement in Crow's feet and found it to be significantly improved ($p < 0.05$) at 4 weeks after using the product compared with that before using the product ($p < 0.05$). Significant improvement continued even after 1 week of discontinuation of the product ($p < 0.05$). There was no significant difference in Crow's feet after 4 weeks of product use and 1 week after discontinuation of the product (Fig 4A and 4B, S1 Table).

**Analysis of skin pigmentation.** We conducted a study on 20 subjects to evaluate the improvement in pigmentation upon using the product. The skin color ($L^*$ value) of the pigmented area was significantly improved ($p < 0.05$) at 4 weeks after using the product compared with that before using the product ($p < 0.05$). Significant improvement continued even after 1 week of discontinuation of the product ($p < 0.05$). There was no significant difference

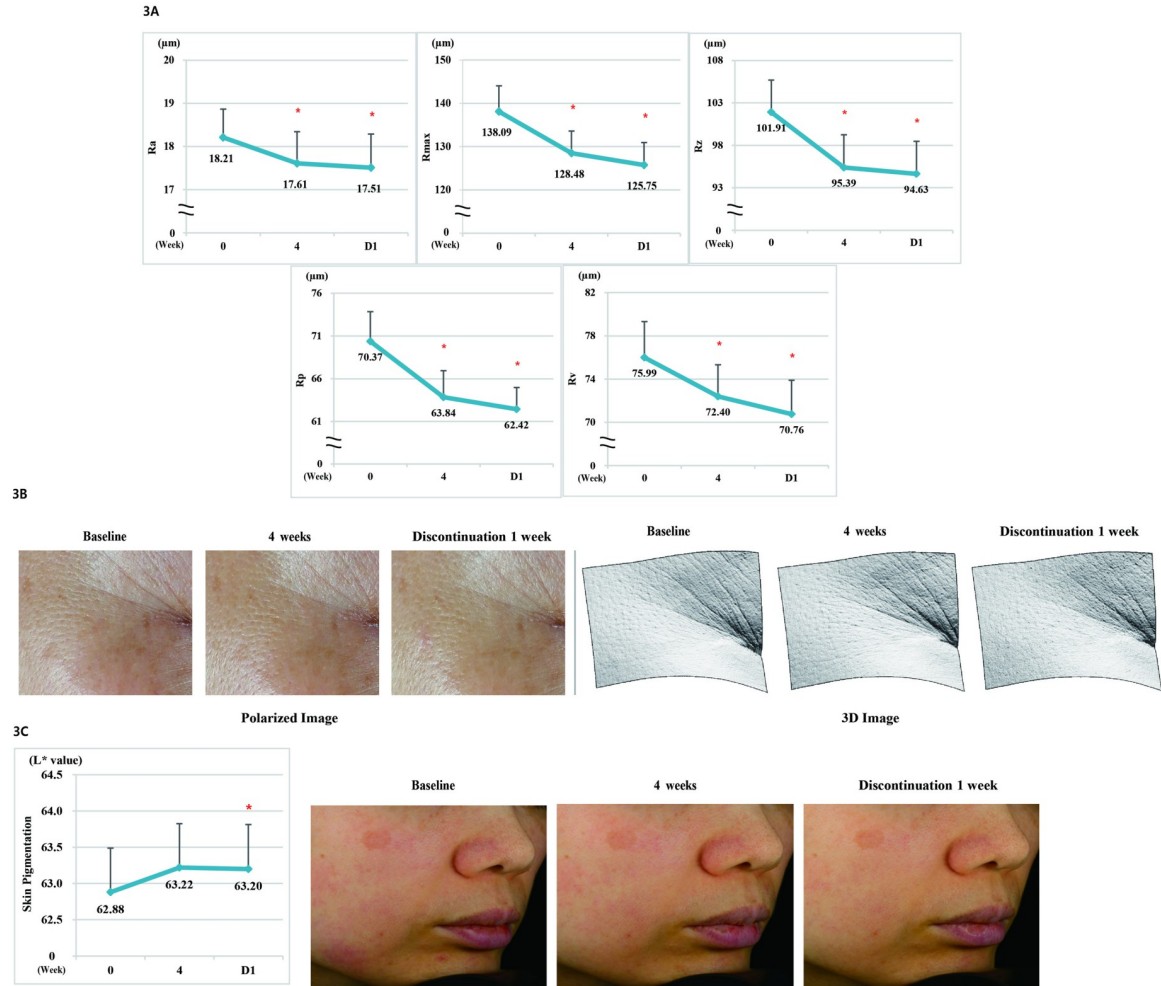

**Fig 3.** (A) Measurement of Average depth of wrinkles, Mean depth biggest wrinkles, Max. depth biggest wrinkles, Total wrinkle count, Total wrinkle volume, Total wrinkle area, Total length of wrinkles, Ra, and Rz parameters. (B) Polarized image and 3D image of Crow's feet, nasolabial folds, forehead wrinkles. (C) F-RAY, the facial contour line image, and graph of the left cheek area. (D) Measurement of skin pigmentation. Cross-polarized image of the left cheek area. (E) Measurement of dermal density and ultrasonographic image of dermal density. (F) Effect for mechanical imprint (pressure) to skin and 3D image mechanical imprint (pressure) relief of the right cheek area.

in the pigmentation after 4 weeks of product use and 1 week after discontinuation of the product use (Fig 4C, S1 Table).

## Analysis of the effects after single application of the product

The results of the clinical trial indicated the changes in fine lines in the eye area, skin sagging, gloss, and hydration immediately after the application of the product once.

**Analysis of fine lines in the eye area.** We confirmed the effect of the product use in improving the fine wrinkles in 20 subjects. When compared with the baseline, all the parameters of fine wrinkles (Ra [Arithmetic average], Rmax [Maximum peak to valley roughness height], Rz [Average maximum height of the profile], Rp [Largest positive deviation], and Rv [Largest negative deviation]) improved significantly immediately after the test product was

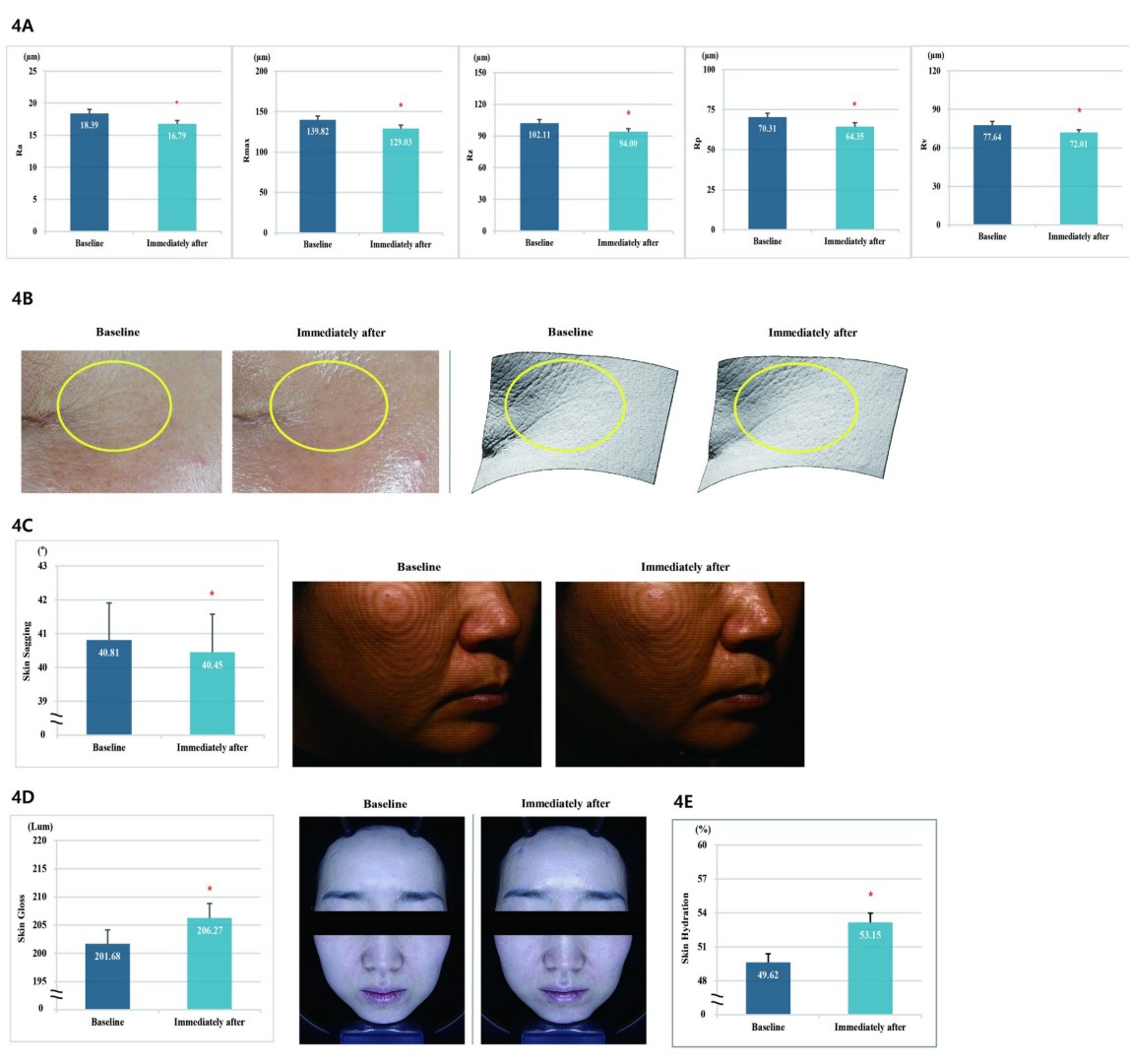

**Fig 4.** (A) Measurement of Ra, Rmax, Rz, Rp, and Rv parameters. (B) Polarized and 3D images of Crow's feet after 4 weeks of product use and 1 week after discontinuation. (C) Effect on skin color improvement after 4 weeks of product use and 1 week after discontinuation and cross-polarized image of skin color in hyperpigmented areas.

applied, with a decrease of 8.70%, 7.72%, 7.94%, 8.48%, and 7.25%, respectively (Fig 5A and 5B, S1 Table).

**Analysis of skin sagging.** We checked the facial lifting effects of the product in 20 subjects. When compared with the baseline, skin sagging was significantly improved ($p < 0.05$) immediately after the test product was applied, with a 0.88% change (Fig 5C, S1 Table).

**Analysis of skin gloss.** We confirmed the skin radiance improvement effect of the product in 20 subjects. When compared with the baseline, the skin gloss improved significantly after the application of the product ($p < 0.05$), with an increase of 2.28% (Fig 5D, S1 Table).

**Analysis of skin surface to a depth of 2.5 mm hydration.** We evaluated the effect of the product in improving the skin surface to a depth of 2.5 mm hydration in 20 subjects. When compared with the baseline, the skin surface to a depth of 2.5 mm hydration was significantly improved ($p < 0.05$) at immediately after the application of the test product, with a 7.11% change (Fig 5E, S1 Table).

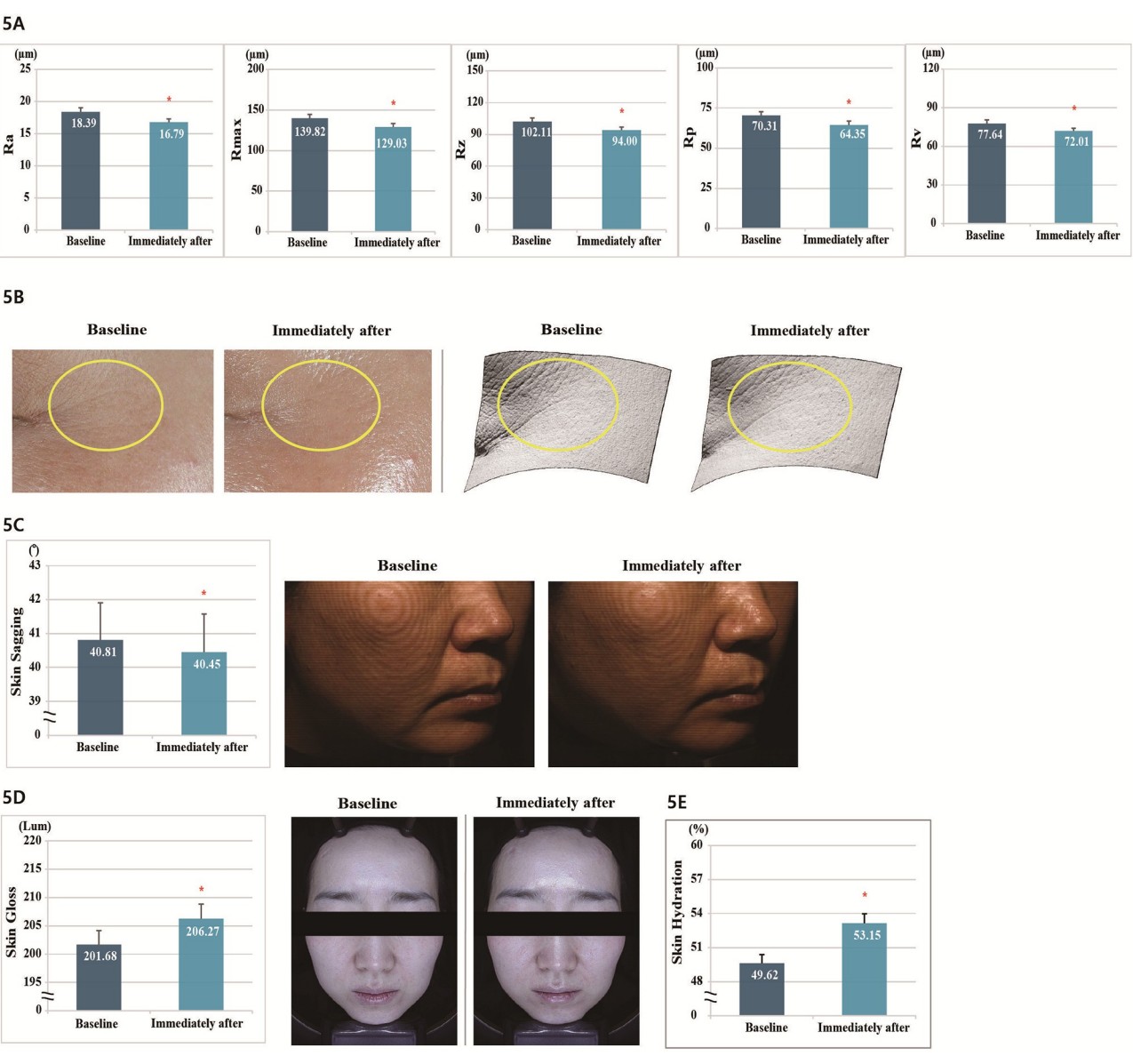

**Fig 5.** (A) Measurement of Ra, Rmax, Rz, Rp, and Rv parameters for fine wrinkles around the eyes. (B) Polarized and 3D images of wrinkles around the eyes after one use of the product. (C) F-RAY the facial contour line image of the right cheek area after using the product once. (D) Effect of a single use of the product in improving the facial gloss and polarized image of skin gloss after a single use of the product. (E) Facial moisturizing effect after using the product once.

**Safety evaluation.** None of the subjects showed any adverse skin reaction during the study period. We performed a 24-hours occlusive patch test. After removing the patch, each test region was observed after 30 minutes and 24 hours, according to the PCPC guidelines. With regard to the primary irritation potential for human skin, the test product was found to be in the "none to slight" range (range of response = 0.00, S1 Text, S2 Table).

## Discussion

The purpose of this study was to evaluate the antiaging potential of a mixture of collagen and ascorbic acid. We investigated whether the synergistic effect of these two substances could be

achieved while maintaining the absorption of collagen (Naticol®) and the stability of ascorbic acid. We found that the product was effective in alleviating skin aging through *ex vivo* and clinical trials.

First, in the evaluation of the permeability of the product across skin stratum corneum in *ex vivo* human skin tissue, fluorescence was evident as a result of penetration, confirming the penetrability of the product down to the stratum corneum of the skin.

In the clinical trial, values for all the parameters for the eye area, nasolabial folds, and forehead wrinkles were significantly improved after 2 and 4 weeks of using the product, and facial contours and skin color were also significantly improved. Moreover, there was a significant improvement in the skin density and mechanical imprint relief 4 weeks after using the product. Mechanical imprint relief is a new clinical method devised on its own. With the decrease in skin elasticity, marks left on the skin do not disappear easily. It was designed mainly from the traces of the duvet or pillow that occur while sleeping. The fact that the marks left upon waking up disappear in a short time can be interpreted as an improvement in the elasticity of the skin. We evaluated the degree of improvement in skin elasticity by creating marks with the application of constant pressure for a certain period of time.

Skin wrinkles and skin color were measured again after discontinuation of the product use following 1 week of its application. Based on the results, it was confirmed that the effect of the product was sustained even after discontinuation. Moreover, it was confirmed that the skin wrinkles, skin lifting, radiance and moisture content in the skin were significantly improved after a single use of the product.

Collagen, which occupies most of the dermal extracellular matrix and plays an important role in maintaining the strength and viscoelasticity of the skin, is a major structural protein accounting for 75% of the dry weight of the skin and maintains its elasticity [25]. Therefore, when collagen is insufficient, the skin loses elasticity and moisture and wrinkles appear [26]. The synthesis of collagen types I and III is affected by ascorbic acid, iron, and silicon, and proline and lysine, the major constituent amino acids of collagen, are through to be involved in the process [20]. In cosmetics, delivery of collagen to the skin is an important challenge. Considering this issue, we selected Weishardt's fish collagen. This collagen, which has low molecular weight (Mean Mw: 2kDa) and high dispersability in a microgranulated form, is a component of TEENIALL [2].

Ascorbic acid, also called vitamin C, is the most powerful and broad-spectrum antioxidant among the antioxidant vitamins present in cells [27,28]. It may not be efficiently delivered to the skin due to its unstable state in cosmetics. In particular, natural ascorbic acid is easily decomposed by oxygen and light [29]. To overcome these shortcomings, we used containers that allowed physical separation of ascorbic acid and collagen. The two substances were mixed immediately before using the product.

## Conclusions

The combined use of ascorbic acid and collagen resulted in skin whitening and wrinkle improvement, as well as in the improvement of skin elasticity, radiance, and moisturizing effects. In addition, the effects of ascorbic acid and collagen were synergistic and were maintained even after discontinuation of the product. Although the effects of the combination of collagen and ascorbic acid were clinically confirmed, the molecular mechanisms underlying the effects of each were not confirmed. Therefore, it would be good to check the effects of each component separately to elucidate the exact mechanism. Moreover, it may be necessary to conduct a study comparing the data for each. We believe that there will be better results if the improvement effect is measured when collagen and ascorbic acid are used over a long time.

## Supporting information

**S1 Checklist. TREND statement checklist.**
(PDF)

**S1 Table. *P*-Values for the graphs included in the figures.**
(DOCX)

**S2 Table. Formula for safety evaluation.**
(DOCX)

**S1 Text. Skin primary irritation procedure.**
(DOCX)

**S1 File.**
(DOCX)

**S2 File.**
(DOCX)

## Author Contributions

**Conceptualization:** Dong Keon Yon, Soo Yun Lee, Go Woon Choi, Da Som Jeon, Bo Bae Oh, So Min Kang.

**Data curation:** Da Yeong Nam, Soo Yun Lee, Byung Ho Shin, Go Woon Choi, Da Som Jeon, Bo Bae Oh, Ji Hyun Kim, Young Yoon, Hyun Jeong Kim, So Min Kang.

**Formal analysis:** Go Woon Choi, So Min Kang.

**Investigation:** Tae Kyeong Ryu, Hanna Lee, Soo Yun Lee, Go Woon Choi.

**Methodology:** Da Yeong Nam, Soo Yun Lee, Go Woon Choi, Bo Bae Oh, Christelle Bruno-Bonnet.

**Project administration:** Soo Yun Lee, Go Woon Choi, Da Som Jeon, Bo Bae Oh, So Min Kang.

**Resources:** Luc Duteil.

**Supervision:** So Min Kang.

**Validation:** So Min Kang.

**Visualization:** So Min Kang.

**Writing – original draft:** Tae Kyeong Ryu, Hanna Lee, So Min Kang.

**Writing – review & editing:** Dong Keon Yon, Byung Ho Shin, Chan Yeong Heo, So Min Kang.

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
