## [Decision Letter · Decision Letter 0]

10 May 2022

PONE-D-22-07546The anti-aging effects of the product including both collagen and ascorbic acid: in vitro, ex vivo and pre-post intervention clinical trialsPLOS ONE

Dear Dr. Kang,

Thank you for submitting your manuscript to PLOS ONE. After careful consideration, we feel that it has merit but does not fully meet PLOS ONE’s publication criteria as it currently stands. Therefore, we invite you to submit a revised version of the manuscript that addresses the points raised during the review process.

We look forward to receiving your revised manuscript.

Kind regards,

Parasuraman Padmanabhan, Ph.D

Academic Editor

PLOS ONE

Journal Requirements:

4. Please amend the manuscript submission data (via Edit Submission) to include authors Tae Kyeong Ryu, Han Na Lee, Da Yeong Nam, Soo Yun Lee, Byung Ho Shin, Go Woon Choi, Da Som Jeon, Bo Bae Oh, Ji Hyun Kim, Young Yoon, Hyun Jeong Kim, Luc Duteil, Christelle Bruno-Bonnet and Chan Yeong Heo.

Additional Editor Comments:

Reproducibility of In vitro & Ex vivo experiments will be repeated and new results may be added.

Reviewers' comments:

Reviewer's Responses to Questions

**Comments to the Author**

1. Is the manuscript technically sound, and do the data support the conclusions?

Reviewer #1: No

Reviewer #2: Partly

Reviewer #3: Yes

2. Has the statistical analysis been performed appropriately and rigorously? 

Reviewer #1: Yes

Reviewer #2: Yes

Reviewer #3: Yes

3. Have the authors made all data underlying the findings in their manuscript fully available?

Reviewer #1: Yes

Reviewer #2: Yes

Reviewer #3: Yes

4. Is the manuscript presented in an intelligible fashion and written in standard English?

Reviewer #1: Yes

Reviewer #2: Yes

Reviewer #3: Yes

5. Review Comments to the Author

Reviewer #1: A clinical trial was conducted which aimed to study the effect of TEENIALL on improving skin wrinkles, skin lifting, and pigmentation areas after 2 and 4 weeks of using the product. Skin wrinkling, sagging, and pigmentation decreased after TEENIALL usage.

Major revisions:

Provide summary tables of results. Include precise p-values instead of “p<0.05” or “p< 0.001.”

Minor revisions:

1- The standard statistical term for average is mean.

2- Study procedure: State precisely the total number of subjects enrolled and analyzed instead of vaguely indicating “over 20 subjects.”

3- Safety Evaluation: Provide an interpretation for “Range of response = 0.00.”

4- State and justify the study’s target sample size with a pre-study statistical power calculation. The power calculation should include: sample size, alpha level (indicating one or two-sided), minimal detectable difference and statistical testing method.

5- To assist in the review process, add line numbering to the document.

Reviewer #2: Reproducibility of In vitro & Ex vivo is not easy as the given details of the methods used/adopted is not sufficient for a scientific journal and its readers.

The source of the cell lines are not given.

Use of generic names for the evaluated products/ details of components could make it of better use for the scientific community to advance science.

Reviewer #3: The manuscript “The anti-aging effects of the product including both collagen and ascorbic acid: in vitro, ex vivo and pre-post intervention clinical trials” by Ryu et.al. elaborates the use of TEENIALL to enhance the synergetic effect between collagen and ascorbic acid, tested to decrease skin wrinkles and hyperpigmentation in humans within 40 to 59 years. The research had focused on use of TEENIALL in reducing aging effects, with visible results after the first use. These findings should thus motivate further research and would deserve publication. However, I do not think that this paper is publishable in its present form and that at least one round of minor revision is needed.

In the introduction, page 2, the sentence starting with “Ascorbic acid is a representative….” can be given as a separate paragraph.

The products Naticol ® and Quali®-C, are mentioned as “raw materials” in most of the places. Please change that to “products” or any other word of similar meaning.

In the study procedure the sentences” In this study, treatment was applied once, 2 and 4 weeks, the study was conducted as a single test. Over 20 subjects meeting the inclusion and exclusion criteria were enrolled” and “For 4 weeks, the product was applied to subjects on a daily basis around the whole face after washing the face twice a day” are unclear and misleading.

Please correct the grammatical errors in the manuscript. There are several grammatical errors found.

The English language is generally unsatisfactory.

I suggest rewriting the conclusion. The line starting with “In summary….can be corrected”

6. PLOS authors have the option to publish the peer review history of their article (what does this mean?). If published, this will include your full peer review and any attached files.

Reviewer #1: No

Reviewer #2: No

Reviewer #3: No

---

## [Author Response · Author response to Decision Letter 0]

30 Jun 2022

June 13, 2022

Dr. Parasuraman Padmanabhan

Academic Editor

PLOS ONE

Dear Dr. Padmanabhan:

It is with great pleasure that we resubmit the revised version of our manuscript, presently titled “The antiaging effects of a product containing collagen and ascorbic acid: In vitro, ex vivo, and pre-post intervention clinical trial,” by Tae Kyeong Ryu et al. to PLOS ONE. The manuscript ID is PONE-D-22-07546R1. 

We appreciate the decision of the Editors to reconsider our manuscript and are grateful to the reviewers for their meticulous comments. We have sincerely endeavored to address each comment and have made appropriate modifications to our manuscript. We acknowledge that our manuscript has benefited from the insightful suggestions. We hope that the revised manuscript addresses all the concerns and would be considered suitable for publication in PLOS ONE. If you have any further queries, please do not hesitate to contact me.

Thank you for your kind consideration of our manuscript. We look forward to a favorable decision.

Sincerely yours,

So Min Kang, Ph.D.

Major changes and additions in the revised manuscript

We have revised the manuscript and included a number of additional values in supplemental tables.

We marked the edited part of the manuscript with yellow highlight.

<Reviewer #1>

[1] Revision (Major) 

Provide summary tables of results. Include precise p-values instead of “p<0.05” or “p<0.001.”

Response: We would like to thank the reviewer for this comment. As suggested, we have provided the exact p-values for the data presented in the graphs as supporting information (S1 Table). 

[2] Revision (Minor)

The standard statistical term for average is mean.

Response: We agree with the reviewer’s contention that “mean” rather than “average” is the standard statistical term. Therefore, as suggested, we have replaced “average” with “mean” throughout the revised manuscript.

[3] Revision (Minor)

Study procedure: State precisely the total number of subjects enrolled and analyzed instead of vaguely indicating “over 20 subjects.”

Response: Thank you for your comment. We would like to submit that we used “over 20 subjects” in accordance with the cosmetic clinical trial method guidelines followed in the Republic of Korea (Ministry of Food and Drug Safety MFDS, Guidelines for Cosmetics Human Application Test and Efficacy Test). 

The number of people who participated in the actual test has been mentioned for the each evaluation item in the Results section. (The number of subjects participating in the trial is underlined.) The number of participating subjects for all evaluations was 21. However, in the evaluation of “after 1 week of discontinuation of product use,” one subject dropped out due to withdrawal of consent, and finally, 20 participants participated. We performed analysis on those 20 subjects. 

[4] Revision (Minor)

Safety Evaluation: Provide an interpretation for “Range of response = 0.00.”

Response: Safety evaluation aimed at evaluating the primary skin irritation potential of the test materials using a primary patch test for the human skin. We performed a 24-hours occlusive patch test. After removing the patch, each test region was observed after 30 minutes and 24 hours, according to the PCPC guidelines. Over 30 subjects were selected according to the recommendations in the MFDS guidelines. We derived the results with reference to Tables 1 and 2 below. Therefore, using this test procedure for the primary irritation potential for human skin, the test product was found to be in the “none to slight” range (range of response = 0.00). We have added this information the Results section (Line 353 on page 15 of the manuscript) and attached Supporting Text file.

[5] Revision (Minor)

State and justify the study’s target sample size with a pre-study statistical power calculation. The power calculation should include: sample size, alpha level (indicating one or two-sided), minimal detectable difference and statistical testing method.

Response: We apologize for insufficient application of statistical techniques for justifying the target sample size and for not introducing our study methodology in detail. We have added information to reinforce our main results (Line 142 on page 6 of the manuscript). We hope that the changes would address the reviewer’s concern.

[6] Revision (Minor)

To assist in the review process, add line numbering to the document.

Response: We would like to thank the reviewer for the comments. We have added line numbers to the Word document.

<Reviewer #2>

[1] Revision

The source of the cell lines are not given.

Response: We would like to thank the reviewer for the comments. We have added the source of the cell lines in the revised manuscript.

[2] Revision

Use of generic names for the evaluated products/ details of components could make it of better use for the scientific community to advance science.

Response: We would like to thank the reviewer for the pertinent suggestion. We have used generic names instead of “TEENIALL” and “Naticol®.” Moreover, we have added a description about the ascorbic acid and collagen mixed component, which we refer to as TEENIALL, in the revised manuscript.

<Reviewer #3>

[1] Revision

In the introduction, page 2, the sentence starting with “Ascorbic acid is a representative….” can be given as a separate paragraph.

Response: We appreciate this suggestion. We have presented the text mentioned by the reviewer in a separate paragraph the Introduction section (Line 80 on page 4 of the manuscript).

[2] Revision

The products Naticol ® and Quali®-C, are mentioned as “raw materials” in most of the places. Please change that to “products” or any other word of similar meaning.

Response: We would like to thank the reviewer for the comment. We have changed “raw materials” to specific terms (“Naticol® and ascorbic acid”).

[3] Revision

In the study procedure the sentences” In this study, treatment was applied once, 2 and 4 weeks, the study was conducted as a single test. Over 20 subjects meeting the inclusion and exclusion criteria were enrolled” and “For 4 weeks, the product was applied to subjects on a daily basis around the whole face after washing the face twice a day” are unclear and misleading.

Response: We apologize for the lack to clarity. We have revised the description and hope that it conveys the intended meaning. 

[4] Revision

Please correct the grammatical errors in the manuscript. There are several grammatical errors found. The English language is generally unsatisfactory. I suggest rewriting the conclusion. The line starting with “In summary….can be corrected”

Response: We acknowledge that there were language issues in the manuscript. In accordance with the reviewer’s comment, we have employed a professional scientific editing service to ensure that the language of the revised manuscript meets the publishable standards.

---

## [Decision Letter · Decision Letter 1]

24 Oct 2022

The antiaging effects of a product containing collagen and ascorbic acid: In vitro, ex vivo, and pre-post intervention clinical trial

PONE-D-22-07546R1

Dear Dr. Kang,

We’re pleased to inform you that your manuscript has been judged scientifically suitable for publication and will be formally accepted for publication once it meets all outstanding technical requirements.

Kind regards,

Parasuraman Padmanabhan, Ph.D

Academic Editor

PLOS ONE

Additional Editor Comments (optional):

Please addressed by the reviewer raised comments. Follow the journal guidelines.

Reviewers' comments:

Reviewer's Responses to Questions

**Comments to the Author**

1. If the authors have adequately addressed your comments raised in a previous round of review and you feel that this manuscript is now acceptable for publication, you may indicate that here to bypass the “Comments to the Author” section, enter your conflict of interest statement in the “Confidential to Editor” section, and submit your "Accept" recommendation.

Reviewer #1: (No Response)

Reviewer #2: All comments have been addressed

2. Is the manuscript technically sound, and do the data support the conclusions?

Reviewer #1: Yes

Reviewer #2: Yes

3. Has the statistical analysis been performed appropriately and rigorously? 

Reviewer #1: Yes

Reviewer #2: Yes

4. Have the authors made all data underlying the findings in their manuscript fully available?

Reviewer #1: Yes

Reviewer #2: Yes

5. Is the manuscript presented in an intelligible fashion and written in standard English?

Reviewer #1: Yes

Reviewer #2: Yes

6. Review Comments to the Author

Reviewer #1: Minor revisions: (Page numbers refer to those in the tracked changes version of revision 1.)

1- Line 165: Indicate the statistical testing method which attains 95% power.

2- In line 176, "product" is misspelled.

3- Line 187: Consider replacing "around" with "over."

4- Line 254: Drop the word "The" from the beginning of the sentence. Replace "using kurtosis and skewness" with "by examining the kurtoisis and skewness."

5- Line 256: Distinguish between the results in which p<0.05 and p<0.001 were considered statistically significant.

6. Within the text, replace "p<0.05” or “p<0.001” with more precise p-values.

Reviewer #2: Line 65-66 - rephrase to avoid redundant use of word `increase'.

Line 156 - `Product' spelling incorrect

Line 250-251 - Verify if the group without UV irradiation is the control instead of BLANK

7. PLOS authors have the option to publish the peer review history of their article (what does this mean?). If published, this will include your full peer review and any attached files.

Reviewer #1: No

Reviewer #2: No

---

## [Editor Report · Acceptance letter]

1 Dec 2022

PONE-D-22-07546R1 

The antiaging effects of a product containing collagen and ascorbic acid: *In vitro*, *ex vivo*, and pre-post intervention clinical trial 

Dear Dr. Kang:

I'm pleased to inform you that your manuscript has been deemed suitable for publication in PLOS ONE. Congratulations! Your manuscript is now with our production department. 

Kind regards, 

on behalf of

Dr. Parasuraman Padmanabhan 

Academic Editor

PLOS ONE